# Efficiency of Priority Queue Architectures in FPGA †

Lukáš Kohútka 

† This paper is an extended version of our paper published in DSD, Kohutka, L.; Nagy, L.; Stopjaková, V. A Novel Hardware-Accelerated Priority Queue for Real-Time Systems, 2018. In Proceedings of the 21st Euromicro Conference on Digital System Design (DSD), Prague, Slovakia, 2018; pp. 46–53. https://doi.org/10.1109/DSD.2018.00023.

**Abstract:** This paper presents a novel SRAM-based architecture of a data structure that represents a set of multiple priority queues that can be implemented in FPGA or ASIC. The proposed architecture is based on shift registers, systolic arrays and SRAM memories. Such architecture, called MultiQueue, is optimized for minimum chip area costs, which leads to lower energy consumption too. The MultiQueue architecture has constant time complexity, constant critical path length and constant latency. Therefore, it is highly predictable and very suitable for real-time systems too. The proposed architecture was verified using a simplified version of UVM and applying millions of instructions with randomly generated input values. Achieved FPGA synthesis results are presented and discussed. These results show significant savings in FPGA Look-Up Tables consumption in comparison to existing solutions. More than 63% of Look-Up Tables can be saved using the MultiQueue architecture instead of the existing priority queues.

**Keywords:** priority queue; architecture; efficiency; FPGA; SRAM; data sorting; MultiQueue





## 1. Introduction and Motivation

Priority queues (also known as min/max queues) are more popular for their implementation in software. Despite software implementations often being sufficient for many applications, in some cases it is necessary to either sort/queue more data at a faster rate or guarantee a constant response time. Real-time systems and cyber-physical systems demand constant response times and high throughput. Real-time systems are a type of embedded system that involves real-time interaction with the environment. Despite the highest possible performance of a controller for real-time tasks, there is still no guarantee that these tasks will always succeed. A real-time task's success depends not just on the computation result, but also on its completion time. Therefore, a dedicated hardware-accelerated design of the needed real-time functionality is typically needed for real-time systems. The constant latency of all operations in the system, including data sorting and priority queues, is very important for more deterministic and reliable scheduling in hard real-time systems. In such cases, software implementations of data sorting and priority queues cannot operate in constant time with respect to the amount of data to be sorted. An alternative to software implementation is hardware acceleration, which is accomplished by building the priority queues and data sorting into an integrated circuit (e.g., ASIC or FPGA) [1,2].

In the past, several hardware architectures have been developed for data sorting or prioritizing information, but they are largely limited by their logical resources; especially when multiple such priority queues are required, they may have overly high chip area costs in ASIC and high LUTs consumption in FPGA technologies [2–13].

The priority queues are needed for many various reasons in many various applications. One popular usage is task scheduling in operating systems, especially implementation of EDF algorithm and priority-based scheduling [1–3,10–19]. Another popular usage of

priority queues accelerated by hardware (i.e., implemented in ASIC/FPGA) is in the domain of networking, mainly packet scheduling and switching [5–8]. Priority queues are very popular in CPU designs too, they are needed for instruction scheduling within out-of-order execution logic [20] or in memory management units [21] too. Another possible application of priority queues is for accelerated solutions of graph theory problems, e.g., for Dijkstra's algorithm or others [22–28]. Priority queues are also used intensively in network devices such as Ethernet switches designed for distributed real-time systems [29,30].

This paper is focused on efficiency analysis and comparison of existing priority queue architectures implemented in FPGA as well as proposing a new architecture, with a goal to reduce the amount of logic resources needed to implement several priority queues. The main goal is to design a new architecture for implementation of multiple priority queues with reduced resource costs, while maintaining the same performance and constant response time of the queues as long as only one priority queue needs to be accessed at a time. This new hardware architecture, called MultiQueue, is proposed and a comparison with other existing architectures is presented in this paper.

This paper uses the following structure. Section 2 provides a detailed description of all requirements for priority queues implemented in hardware. In Section 3, existing solutions of priority queues are described. The novel MultiQueue architecture is presented in Section 4. Section 5 contains a design verification of the proposed solution as well as existing solutions, which was achieved by simulations in simulator. In Section 6, FPGA synthesis results are presented and compared. The last section summarizes the achieved research results.

This paper is based on our previous work published in *DSD* 2018 [17], with substantial extensions.

## 2. Priority Queue Requirements

Research presented in this paper is focused on priority queues performing data sorting in real-time systems, i.e., queues that prioritize top-priority items. This means that the goal is to create either a min queue or max queue that maintains items by sorting them.

Priority queue is a data structure that can store and sort some data that are called items. Each item consists of three parts: *SORT_DATA*, *ID* and *VALID*. The *SORT_DATA* represents the value that is used to sort the items in the queue. Output of min queue should be the item with the minimum *SORT_DATA* value. Output of max queue should be the item with the maximum *SORT_DATA* value. The *ID* represents either a unique identification number of the item or payload data that are relevant to the application, but not relevant for the sorting within the priority queue. The payload has no impact on the sorting decisions. This value can also serve as a pointer/address to memory. The *VALID* part is a 1-bit value indicating whether the item is valid or empty. Logic 1 means that the item is valid. Logic 0 means that the item is invalid, i.e., the given part of the queue is empty. If the output of the queue is not valid, then this means that the whole queue is empty. The *VALID* field is optional, as it can be implicitly hidden in *SORT_DATA* or *ID*, which can use a special value for invalid/empty entries.

The operation of the priority queue is controlled by providing instructions to the interface of the queue. Priority queues are expected to support at least two basic operations: ability to insert a new item into priority queue according to item priority/sorting value (instruction INSERT), and ability to remove an existing item from priority queue (instruction REMOVE). Apart from that, it is expected that there are clock cycles when neither of these operations are used. Therefore, an instruction NOP is needed too. Furthermore, if the priority queue is used very intensively (i.e., if high throughput is required), then it is beneficial if the priority queue supports simultaneous (parallel) execution of both operations, item insertion and item removal—we call this instruction INSREM. Table 1 shows the list of instructions that are usually provided by priority queues.

**Table 1.** Priority queue instructions.

| Name | Opcode | Description |
|---|---|---|
| NOP | 00 | It is used to keep the accelerator idle. There is no operation performed. The queue output remains unchanged. This instruction is required. |
| INSERT | 01 | An item is added to the priority queue. It is inserted in a way that maintains the items in the queue sorted. This ensures that the item with the lowest/highest *SORT_DATA* value remains as the item that is retrieved from the priority queue. This instruction is required. |
| REMOVE | 10 | Among the items in the queue, one item is removed according to item *ID*. The remaining items are rearranged so that the top priority item is the output of the queue. This instruction is required. |
| INSREM | 11 | One item is removed from the queue and simultaneously, one item is inserted into the queue. This is a combination of INSERT and REMOVE instructions. This feature is not required, but optional. In other words, this serves just as an optimization of performance of the priority queue so that both INSERT and REMOVE can be performed in parallel. |

Priority queues typically have some parameters defined too. These parameters can be set in the RTL code before compilation and synthesis of the design:

- *MAX*—a 1-bit value indicating whether a queue is the minimum queue or the maximum queue.
- *SD_W*—this specifies how many bits are used for *SORT_DATA* representation, which is determined by the range of values to be sorted, for example if *SD_W* = 10, then the range is 1024 possible values.
- *ID_W*—an integer number defining the width of *ID* values. This indicates how many bits should be used to represent an *ID* value.
- *SIGNED*—indicates whether the *SORT_DATA* values are signed or unsigned.
- *CAPACITY*—the maximum number of items that can be accommodated in a priority queue, i.e., the capacity of one priority queue.

The most important design attributes of HW-accelerated priority queue (i.e., implemented in ASIC or FPGA instead of software) are the following:

- Constant response time—time interval between the start of instruction and the updated output of priority queue. Constant time complexity is required, which means all instructions must provide output in a constant number of clock cycles. By constant response time, we mean that the clock cycles number does not change regardless of how many items are currently stored in the queue and regardless of queue capacity. Constant response times contribute to better overall predictability and determinism of the whole real-time system [1].
- High throughput. Based on the clock frequency multiplied by the number of clock cycles required to use one instruction, this attribute is determined. Within the same clock domain, clock frequency is dependent on critical path lengths of the accelerator as well as critical paths of other components of the system. No substantial advantage can be gained from achieving a significantly shorter critical path length of the priority queue than the rest of the clock domain, but the critical path of the priority queue should not be longer than the rest of the clock domain either. Using priority queue instructions should take as low a number of clock cycles as possible—the lower, the better.

- Low area and energy cost. The implementation technology determines this attribute. FPGAs are evaluated by the number of logic resources (e.g., LUTs, registers and RAM bits) based on their device selection. ASICs are evaluated by the number of transistors or by the dimensions of the manufactured chip.

If these parameters are applied at their minimum values, the attributes listed above can be efficiently optimized (depending on the application requirements): *ID_W*, *SD_W* and *CAPACITY*. Furthermore, these attributes are highly influenced by the priority queue architecture. The goal of this study is therefore to reduce resource costs by developing even more efficient architecture than the existing architectures.

## 3. Related Work

Many architectures exist for sorting data in priority queues while keeping instruction response times constant. FIFO with MUX Trees [2,3], Shift Registers [4,5], DP RAM Heapsort [6], Systolic Array [8,9], Rocket Queue [16], and Heap Queue [17] are among the most popular architectures.

In the FIFO approach, the complexity of the MUX Tree part contributes to inefficiency due to a long critical path when higher capacity is selected. It also takes up an excessive amount of chip area when high capacity is selected [2,3].

A more efficient approach than the previous one is the Shift Registers architecture, but there is still a problem with the critical path length. Each cell in the Shift Registers architecture consists of a comparator, control logic and a set of registers for storing one item. In the queue, all cells receive the same instructions simultaneously from the input (they are connected within the same line), so they can exchange items with their neighbors. Since an increasing number of cells in the queue leads to a longer critical path, this architecture is only suitable for small capacities due to the wide bus width that is available for simultaneously sending instructions to all cells, and because of the exchange of control signals between them. Figure 1 illustrates a four-cell example of Shift Registers architecture.

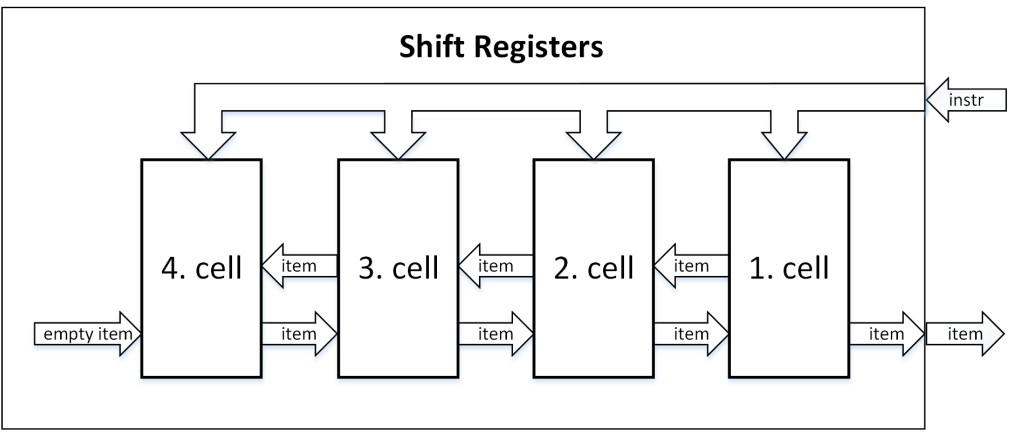

**Figure 1.** Shift Registers architecture [4,5].

In DP RAM Heapsort, items are stored within a dual-port RAM, which makes it a relatively efficient sorting architecture. In spite of this, it is not possible to perform INSERT or POP instructions separately. This architecture can only handle POP and INSERT instructions together, and for this reason, it cannot be used to implement priority queues [6].

A Systolic Array is similar to Shift Registers, except that the critical path problem is solved with the use of pipelining. It consists of homogeneous cells that are interconnected within one line. Except for the first and the last cells in the queue, each cell is neighbored by another cell to the left and to the right. The first cell is the only one to supply its output to the output of the whole queue and to receive instructions from the input. In the same way that instructions are propagated through pipeline stages of pipelined CPUs, instructions are gradually propagated from the first cell to the last cell (one cell at a time). While Shift

Registers architecture applies instruction to all cells in parallel via shared bus, the Systolic Array architecture gives the instruction to move sequentially, from the first cell to the last cell, at a speed of one cell per clock cycle. Even though this sequential processing might sound like a performance drawback, the instructions can be pipelined, maintaining high throughput. Since the output of this priority queue is obtained from the first cell, this output is updated already at the first cycle of instruction, providing the same instruction latency as Shift Registers architecture (one clock cycle) [8,9].

Figure 2 illustrates the Systolic Array architecture with four cells. The first cell from the right is the first cell in the queue, also serving as an external interface. Only clock and reset signals are propagated in parallel [8,9].

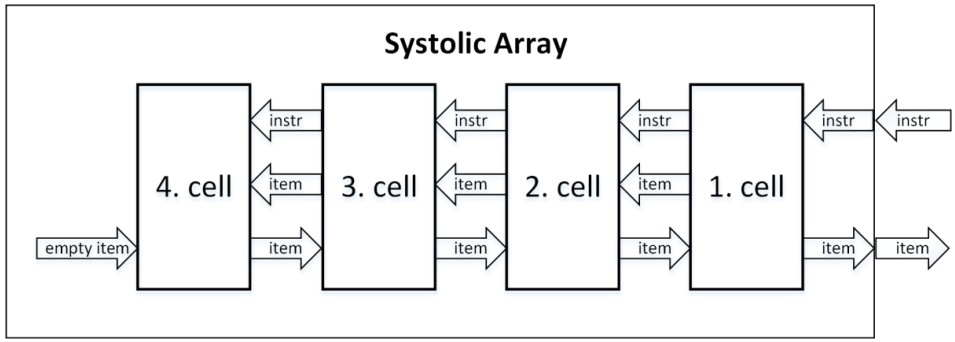

**Figure 2.** Systolic Array architecture [8,9].

In the Systolic Array, each cell represents a pipeline stage, including the pipeline register. A single instruction propagates through the whole structure in N clock cycles, where N refers to the total number of cells. Pipelining has the effect of executing different instructions simultaneously in each cell due to its design. To read an updated output of the queue, it takes only two clock cycles (one clock cycle to update the first cell and one clock cycle to read from the updated cell) since the output of the queue is already updated at the beginning. In this architecture, a new instruction can be added to the priority queue every two clock cycles, which means the instruction response time for this architecture is always two [8,9].

A newer hardware architecture of a priority queue, the Rocket Queue, is presented in [16]. Using the Rocket Queue architecture instead of the Systolic Array architecture can save over 41% of logic resources.

Rocket Queue is an inherited architecture that evolves from Systolic Array and DP RAM Heapsort. Rocket Queue consists of levels, which have two types: duplicating and merged levels. Figure 3 illustrates the Rocket Queue architecture using 11 merged levels and 3 duplicate levels. It is possible to increase the number of duplicating levels, but more than five duplicating levels are not recommended because the critical path would become too long [16].

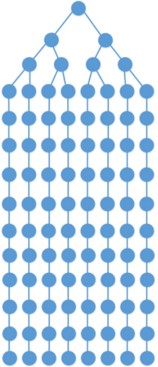

**Figure 3.** Rocket Queue with three duplicating levels [16].

Comparator logic consumes a large amount of resources in the queues. Instead of using one comparator per cell (per item), Rocket Queue uses a single common comparator for all cells within a level instead. So the Rocket Queue architecture has fewer resource costs due to the number of comparators being determined by the number of levels, not the number of cells [16].

As another priority queue architecture, Heap Queue can also be considered. Figure 4 shows that Heap Queue is layered into levels, in a manner similar to Rocket Queue. Each level consists of a Control Unit (CU) and a different number of Item Storage blocks (IS). In total, the IS can be used to preserve one item and one number, which is used to maintain tree balance. Each successive level contains twice as many IS blocks as the previous one. In the first three levels, IS blocks are implemented using registers due to the small size of these memories. In all other levels, IS blocks are implemented using Dual-Port RAM memories, which are much more area-efficient. As an interface for the whole queue, the Control Unit of the first level provides its item as an output, which represents the item with the top priority among all items in the queue [17].

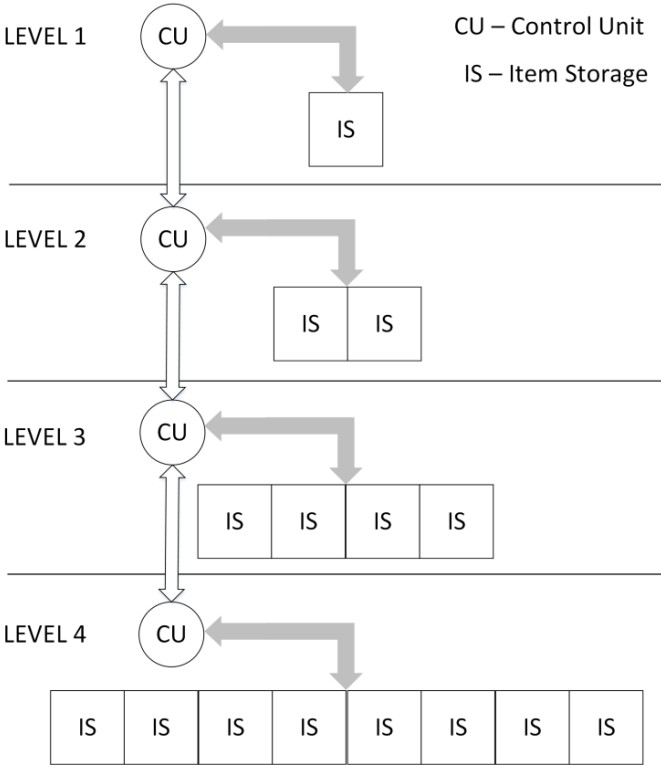

**Figure 4.** Heap Queue with first four levels [17].

Every priority queue architecture has some advantages and disadvantages. The Shift Registers architecture is the simplest, but the frequency does not scale well with increasing numbers of items. Systolic Array solves the timing problem of Shift Registers by adding pipeline registers into the design, making this architecture the most expensive in terms of logic resources or chip area. Although Rocket Queue consumes fewer resources than the previous two architectures, it is also relatively complicated to implement and also needs to use true dual-port SRAM memories. When a large number of items must be held in the queue, the Heap Queue architecture is the most resource-efficient to implement. On the other hand, the Heap Queue is able to remove only the top-priority item from the queue, not any item based on item *ID*, which is often required from many applications. These priority queues are all highly efficient in terms of performance, requiring one or two clock cycles per instruction, regardless of the current number of items in the queue or even capacity of the queue [16,17].

## 4. Proposed MultiQueue

We propose a new architecture, called MultiQueue, which can be used for implementation of multiple priority queues. This architecture is based on existing architectures, Shift Registers and Systolic Array. The items within the queues are stored in SRAM memory instead of registers. The main idea behind MultiQueue architecture is that all combinational logic that is used in Shift Registers and Systolic Array is shared for multiple queues under a condition that only one queue is accessed with an instruction at a time. This way, the combinational logic is used for one queue only. All other queues are idle and do not need the combinational logic. Thanks to the sharing of combinational logic for multiple queues, a significant amount of logic can be saved, which should lead to significant chip area and energy savings.

Figure 5 represents a standard approach for implementation of multiple priority queues, where N individual priority queues are instantiated and multiplexed. This is a typical way to provide a sufficient number of priority queues depending on system requirements. Each priority queue works independently from the other queues and any architecture for implementation of the priority queues can be used, e.g., Shift Registers or Systolic Array, where one register represents data storage for one item, the same as in Figures 1 and 2. The MUX displayed in Figure 5 is used to select one of the priority queues for actual usage, be it an insertion of a new item into one of the queues, removal of item from one queue or reading the output of one queue. The selection criterion depends on application needs/requirements.

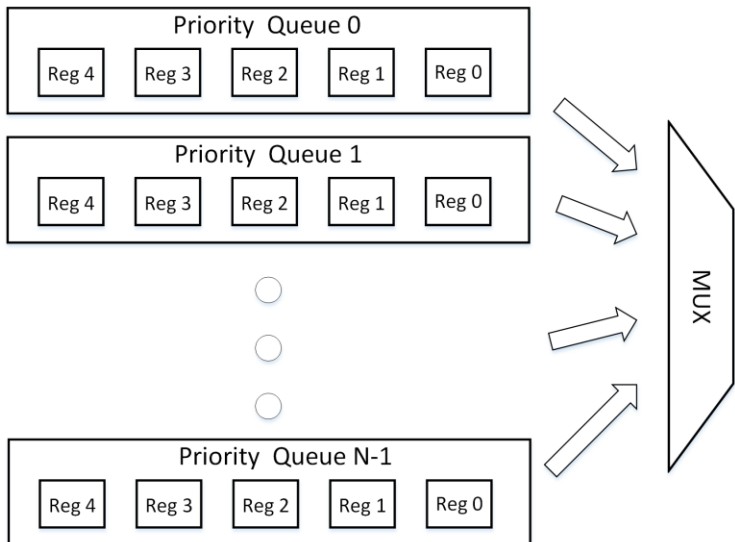

**Figure 5.** Multiple priority queues with multiplexed output.

The problem of the above mentioned approach is that the combinational logic of the priority queues is wasted because only one priority queue is used at a time, leaving the rest of the queues unused. In order to avoid this waste of resources, we propose a new architecture—MultiQueue, which minimizes the amount of combinational logic by sharing the same logic for multiple queues within the same column/register number. This architecture is displayed in Figure 6. In this example, there are N priority queues, each with a capacity of five items (cells 0 to 4). The MultiQueue architecture uses SRAM memories instead of registers. While the standard approach with multiple priority queues needs M registers for each queue, where M is the priority queue capacity, the proposed MultiQueue uses M SRAM memories instead. Each SRAM memory is used for one column, i.e., for a capacity of one item for all priority queues. Each column is controlled by one Sorting Cell module. The logic of these Sorting Cell modules is almost identical to Systolic Array Cells, except for the usage of SRAM for storing items (via standard single-port SRAM interface) instead of using a register inside the cell. Additionally, apart from Systolic Array

architecture, where the instruction is executed in one cell per clock cycle, the MultiQueue architecture is able to process the instruction in multiple cells in parallel (the same way as in Shift Registers architecture) per clock cycle. This number of cells per clock cycle is a parameter that can be set according to critical path and clock frequency requirements—to avoid negative slack time in Static Timing Analysis (STA) on one hand but to maximize the number of cells used in parallel on the other hand, which minimizes the amount of pipeline registers. The SRAM memories must have the memory depth set to the number, which is equal to the number of priority queues (i.e., SRAM depth = N, where N is the number of priority queues in MultiQueue).

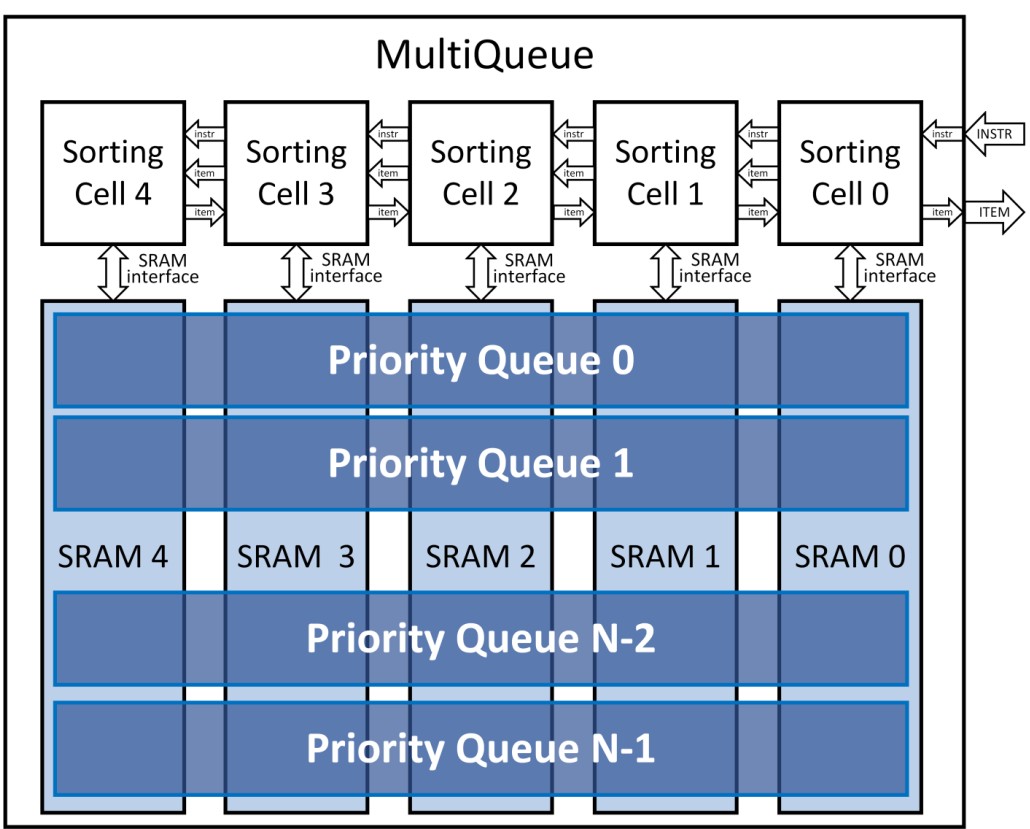

**Figure 6.** MultiQueue architecture.

The total amount of registers in the original approach is N. M (i.e., number of priority queues multiplied by the capacity of one queue). The proposed MultiQueue contains M SRAM memories, each with depth of N. Since SRAM memories are usually implemented as memories with depth set as a power of 2, the MultiQueue architecture can be used for systems with 16, 32, 64, etc. priority queues only. This is the main disadvantage of the proposed solution. Additionally, we assumed that all priority queues have the same queue capacity, which might not be true for some applications. In that case, the priority queue with the highest queue capacity defines the number of SRAM memories in MultiQueue, leading to suboptimal solutions. If the priority queues are supposed to be very long, this means that a number of SRAM memories will be needed. These SRAMs are, however, still much less expensive than the number of registers (and LUTs) that are needed for the ordinary/original approach without SRAMs.

For each register of the original priority queues, there is one single-port SRAM memory used instead. The depth of each SRAM memory depends on the number of priority queues the MultiQueue is supposed to represent. The SRAM address value depends on the number of priority queues that are supposed to be used for the given instruction. This way, the MultiQueue needs an amount of combination logic of one priority queue only, not N priority queues.

The Sorting Cell represents the control logic for instruction decoding and execution, including the reading and writing from/to SRAM memory. The structure of the Sorting Cell is shown in Figure 7. There are two comparators in the Sorting Cell. One comparator checks whether the *SORT_DATA* of the input item provided by instruction (input_item_ff_SD) is better than the *SORT_DATA* of the internal item (this_item_SD). Another comparator is used for detection, whether the input item provided by instruction (kill_ID_ff) has the same ID as the internal item (this_item_ID). Each instruction takes two clock cycles. The first cycle is used to read the existing internal item from SRAM, which is provided via the sram_dataout input port for the second cycle. The second cycle is used for comparing the internal item with the input item using the abovementioned comparators and making a decision whether we write back to SRAM (sram_we), overwriting the internal item (via sram_datain port), either with the input item (input_item_ff), or with the item from the previous cell (right_item), or with the item from the next cell (left_item).

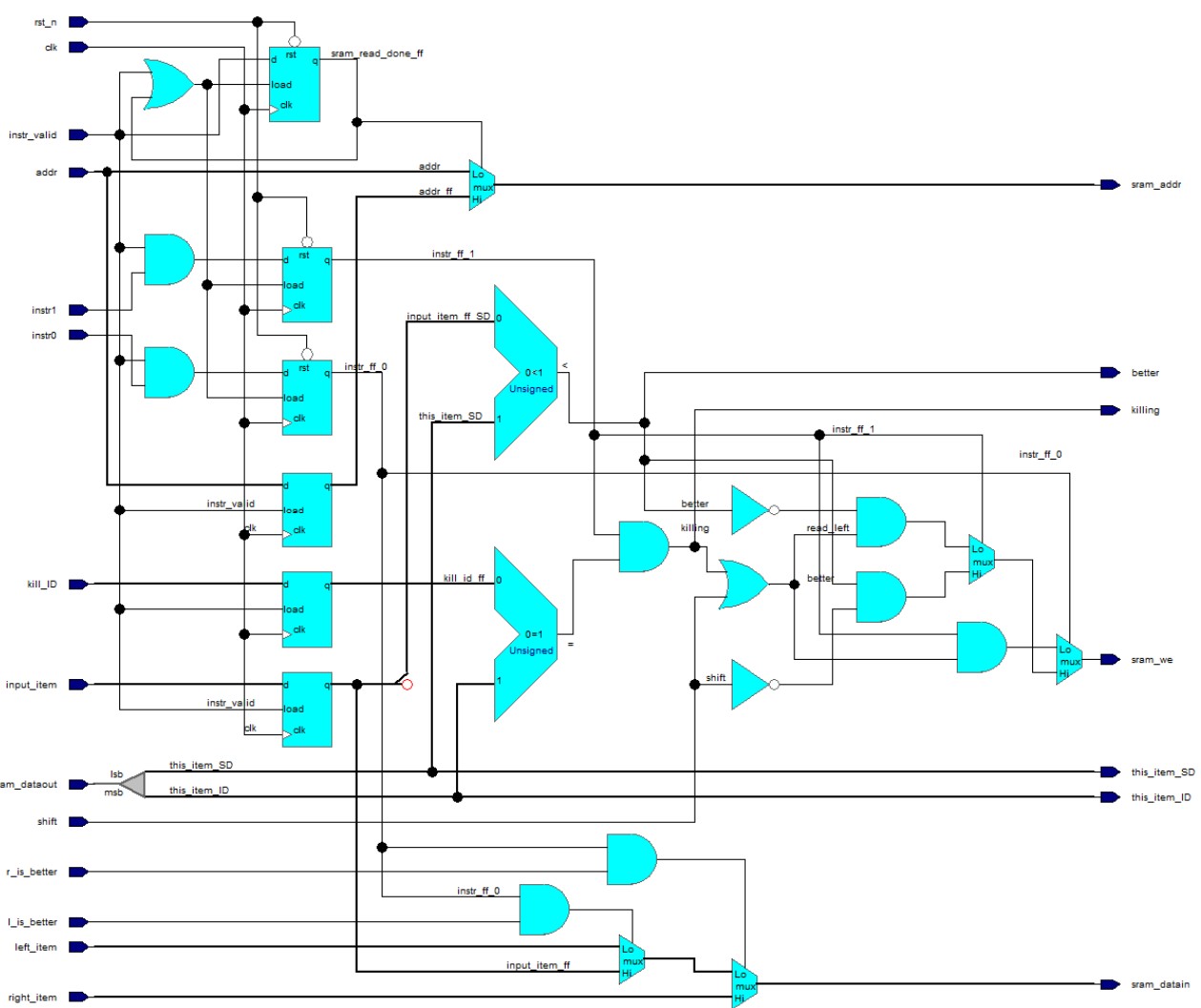

**Figure 7.** Structure of Sorting Cell used in MultiQueue architecture.

The MultiQueue architecture is able to execute any of the required instructions—INSERT, REMOVE, INSREM or NOP, and the execution always takes two clock cycles regardless of actual and/or maximum number of items stored in the queues. However, only one queue can be accessed with the instruction at a time. Using multiple priority queues with the instruction (i.e., writing into the queues) in parallel is not allowed. Because of SRAM usage in this architecture, there is also a requirement for minimum number of priority queues to be used in MultiQueue, which depends on minimum possible depth

of SRAM memories, which depends on the selected implementation technology. For FPGAs, this number is usually smaller than for ASIC technologies. For example, the FPGA Cyclone V that was used for FPGA synthesis experiments in this paper needs at least four priority queues in the MultiQueue because the minimum SRAM depth is 4. Smaller depth would replace the SRAMs with registers, which would lead to less efficient design. ASIC implementations typically have a minimum SRAM depth of 16, 32 or more. Therefore, the MultiQueue architecture is generally more suitable for FPGA implementations.

## 5. Design Verification

The proposed MultiQueue architecture as well as several existing architectures of priority queues was described using SystemVerilog language. Afterwards, correct functionality was verified in simulations. These priority queues were tested as coprocessors that support the instructions listed in Section 2, Priority Queue Requirements. The following priority queues were implemented and verified: Shift Registers, Systolic Array, Rocket Queue, Heap Queue and MultiQueue.

Additionally, for the verification phase, a simplified version of Universal Verification Methodology (UVM) was used. Since priority queue interfaces are relatively simple, UVM's use could also be simplified. We used just one test procedure for generating constrained random inputs, a predictor and a scoreboard to simulate the device under test (DUT). Since UVM transactions are just one instruction performed in two clock cycles, we do not need to use agents for interfacing the DUT. As part of the test procedure, millions of instructions are generated with fixed instruction opcodes and *ID*s, but with randomized *SORT_DATA* values. As the name indicates, the predictor is a module that is responsible for prediction of DUT output based on test inputs (like a DUT, but with higher level of abstraction, just as in high-level software languages). The predictor description is purely sequential, software-like and relatively high level. To order the items in the queue, a SystemVerilog queue structure and sort() function are used. Figure 8 illustrates the entire testbench architecture used for verification.

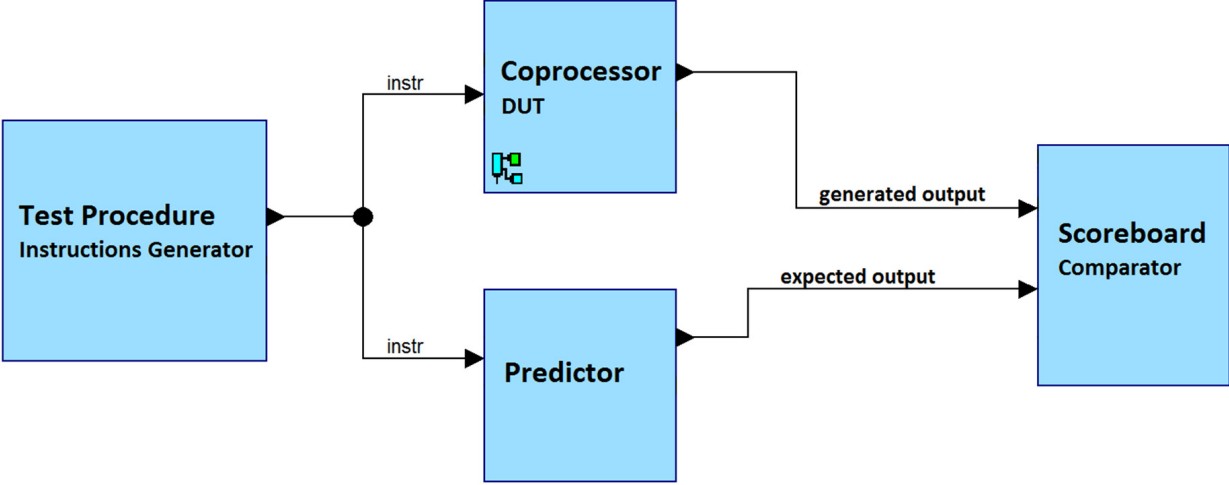

**Figure 8.** Testbench architecture.

All designed queues were verified through 500,000 test iterations, each consisting of 755 instructions generated by the test procedure, with one third being INSERT, one third being REMOVE and the last third being INSREM. In these tests, all queues were loaded to full capacity. Various random configuration parameters were used for the priority queue verification, including: 16 queues, 8bits for item *ID*, 256 items per queue and 32 bits of random *SORT_DATA* values.

## 6. FPGA Synthesis Results

All versions of priority queues listed in the previous section were synthesized into FPGA with a goal to compare their resource consumption efficiency. This synthesis was conducted on an Intel FPGA Cyclone V (5CSEBA6U23I7) operating at 100 MHz. We have conducted a comparison of Adaptive Logic Module (ALM) consumption, which represents the Look-Up consumption, i.e., combinational logic and registers together. In addition to that, SRAM bits consumption was compared too, but only Heap Queue and MultiQueue architectures use this kind of resource. Other architectures are register-based only, thus their consumption of SRAM bits was zero. However, the ALM (LUT) consumption is much more important than the SRAM bits because the ALM resource is usually needed in digital design much more than SRAM bits. Most digital system designs are typically limited by the ALM resource limitations, not SRAM bits.

All priority queues were synthesized for usage with items that consist of 40 bits: 32 bits are used for *SORT_DATA* and 8 bits for *ID*. We decided to use this number of bits as 32 bits are used for standard integer numbers and 8 bits for *ID* is enough for a priority queue capacity of up to 256 items. Of course, these numbers can be adjusted to different values depending on application needs. The synthesis results are presented in Table 2. The first column contains the number of priority queues. The second column is the maximum number of items the priority queue can contain. The third column is the total number of items that can be stored in all priority queues in total, which can be obtained by multiplying the first two columns. The fourth column contains ALM consumption of Shift Registers (ShiftRegs) architecture. The fifth column contains ALM consumption of Systolic Array (SysArray) architecture. Column number six contains ALM consumption of Rocket Queue (RocketQ) architecture. The following two columns contain synthesis results of Heap Queue (HeapQ) architecture. The last two columns contain synthesis results of the proposed MultiQueue (MultiQ) architecture. Since only MultiQueue is specially designed for multiple priority queues, all other priority queues were synthesized individually as a one priority queue only, and then the results were multiplied by 16, 32 or 64 depending on the needed number of priority queues. For example, Shift Registers (ShiftRegs—fourth column from left) with priority queue capacity of 32 items consumed 2315 ALMs and this number was then multiplied by 16 to mimic an implementation of 16 of these priority queues, i.e., 2316 × 16 = 37,040 ALMs.

**Table 2.** FPGA synthesis results of various priority queue architectures.

| Number of Priority Queues | Priority Queue Capacity (Items) | Total Capacity (Items) | ShiftRegs (ALMs) | SysArray (ALMs) | RocketQ (ALMs) | HeapQ (ALMs) | HeapQ (RAM bits) | Proposed MultiQ (ALMs) | Proposed MultiQ (RAM bits) |
|---|---|---|---|---|---|---|---|---|---|
| 16 | 32 | 512 | 37,040 | 43,248 | 36,656 | 31,024 | 36,736 | 2432 | 20,480 |
|  | 64 | 1024 | 74,112 | 87,056 | 63,568 | 37,376 | 79,744 | 4894 | 40,960 |
|  | 128 | 2048 | 158,416 | 174,384 | 115,376 | 45,856 | 165,504 | 9397 | 81,920 |
|  | 256 | 4096 | 321,168 | 349,696 | 218,432 | 53,264 | 337,280 | 19,605 | 163,840 |
| 32 | 32 | 1024 | 74,080 | 86,496 | 73,312 | 62,048 | 73,472 | 2445 | 40,960 |
|  | 64 | 2048 | 148,224 | 174,112 | 127,136 | 74,752 | 159,488 | 4894 | 81,920 |
|  | 128 | 4096 | 316,832 | 348,768 | 230,752 | 91,712 | 331,008 | 9357 | 163,840 |
|  | 256 | 8192 | 642,336 | 699,392 | 436,864 | 106,528 | 674,560 | 19,640 | 327,680 |
| 64 | 32 | 2048 | 148,160 | 172,992 | 146,624 | 124,096 | 146,944 | 2350 | 81,920 |
|  | 64 | 4096 | 296,448 | 348,224 | 254,272 | 149,504 | 318,976 | 4711 | 163,840 |
|  | 128 | 8192 | 633,664 | 697,536 | 461,504 | 183,424 | 662,016 | 9420 | 327,680 |
|  | 256 | 16,384 | 1,284,672 | 1,398,784 | 873,728 | 213,056 | 1,349,120 | 18,808 | 655,360 |

After comparing the results from Table 1, it is clear that the proposed MultiQueue architecture saves a significant number of ALMs (LUTs) in FPGA. The least significant ALM saving was achieved when 16 priority queues were used, each with a size of 256 items, if Heap Queue was used originally. The Heap Queue solution consumed 53,264 ALMs and the proposed MultiQueue consumed 19,605 ALMs, which means a 63.19% reduction of ALM consumption. The most significant ALM saving was achieved when 64 priority

queues were used, each with a size of 256 items, if Systolic Array was used originally. The Systolic Array solution consumed 1,398,784 ALMs and the proposed MultiQueue consumed only 18,808 ALMs, which means 98.66% reduction of ALM consumption.

In terms of performance, all solutions that were discussed and compared in this research paper use a shared interface for execution of one item insertion/deletion/reading from one priority queue at a time. All compared solutions provide the same performance for the application that is using the priority queues. The latency and throughput are two clock cycles per instruction, with the exception of Shift Registers, which is a solution that needs only one clock cycle per instruction, but at a cost of far too long critical paths and poor scaling with priority queue capacity, making such an approach impractical for high-performance applications. The proposed MultiQueue architecture operates with the same performance as a collection of several priority queues based on Heap Queue, Rocket Queue or Systolic Array architecture and is muxed according to Figure 5. For 100 MHz clock and 32-bit items, the performance of all these solutions (Heap Queue, Rocket Queue, Systolic Array and MultiQueue) is 1.6 Gbits/s.

In terms of total energy consumption of all compared solutions, it consists of two parts: static energy consumption (leakage) and dynamic energy consumption. The static energy consumption (leakage) is directly proportional to the chip area costs if the solutions were implemented in ASIC. For FPGA implementations, the static energy consumption depends on FPGA chip selection only. The dynamic energy consumption depends on which priority queue architecture is selected and on input/instructions sequence only. For these reasons, all compared solutions reported the same total energy consumption of 427.3 mW. These results were obtained from Quartus Prime 16.1.0 using default settings of PowerPlay Power Analyzer Tool.

## 7. Conclusions

In this paper, existing priority queue architectures were reviewed and a new architecture for multiple priority queues was presented. This architecture is called MultiQueue and its FPGA implementation can save a significant number of FPGA LUTs in comparison to existing, conventional priority queue solutions. Four existing priority queues plus the new MultiQueue were implemented, tested and synthesized in FPGA to compare their efficiency. The synthesis results show that the proposed MultiQueue architecture can save the majority of combinational logic thanks to the sharing of logic among multiple queues. The more priority queues are implemented within MultiQueue architecture, the bigger the relative reduction of combinational logic that is observed. The only disadvantage and limitation of the MultiQueue architecture is that only one priority queue can be used/accessed at the same time. The number of needed priority queues as well as whether it is required to have access to multiple priority queues in parallel depends on the system requirements and application that will be using these priority queues. All analyzed priority queues including the proposed MultiQueue architecture are able to process any combination of instruction sequences.

While the proposed MultiQueue can save a significant amount of logic resources (LUTs) in FPGA, it is limited for applications that need several priority queues with shared access interface. The number of priority queues can be 16, 32, 64 or more. The proposed solution is not suitable for applications that need access to multiple priority queues simultaneously or applications that use less than 16 priority queues.

**Funding:** This research received no external funding.

**Institutional Review Board Statement:** Not applicable.

**Informed Consent Statement:** Not applicable.

**Data Availability Statement:** Not applicable.

**Conflicts of Interest:** The authors declare no conflict of interest.

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
