# Peer review of "Efficiency of Priority Queue Architectures in FPGA†"

_jlpea, doi:10.3390/jlpea12030039_

Round 1

Reviewer 1 Report

The publication provides an architecture for sharing the logic between multiple priority queues. The presentation of the idea is very poor. In particular, the authors explain in detail obvious things, such as the supported instructions and how they have tested their implementation, while there is no clear explanation of the architecture itself. A detailed block diagram showing how the data/memories are accessed is missing. Figure 6 simply provides a high-level architecture which is not clear. The paper needs major revision and better explanation of the architecture focusing more on implementation details. Also the idea of sharing the logic between multiple queues seems to be obvious. 

Author Response

Dear reviewer, thank you for providing valuable and useful feedback.
The idea of sharing the logic between multiple priority queues may seem to be obvious but it was never published before. The Figure 6 is high-level intentionally - to avoid getting lost in less important details and because this approach is universally applicable to any set of priority queues, regardless of whether they are based on Shift Registers, Systolic Array or any other architecture.

Reviewer 2 Report

Comments 

This work proposes a priority queue hardware accelerator design which aims to improve the energy and resource efficiency over existing architectures, while also maintaining roughly the same throughput and response time. The proposed solution is novel but only addresses the problem provided certain requirements are satisfied. The text is well structured and guides the reader to the point. However, there is some key information missing and some points should be argued further. 

Strengths 

  •  The main strength of this work is the notable reduction in LUT usage, when compared with other architectures.  
  • The main idea of the architecture, which is the shared use of the control logic by all priority queues, provides a fitting solution, provided that the system only accesses one queue at a time.  

Weaknesses 

  • The work promises significant improvement in area and energy efficiency without a tradeoff in response or throughput. The results do indeed show a response time of 2 cc, similarly with other architectures and a significant improvement in resource consumption. However, there is no mention of energy performance and most importantly of throughput.  
  • The MultiQueue architecture is compared against other single-queue architectures by multiplying area usage. However, when used in practice, multiple single-queue instances working in parallel, could potentially provide a throughput advantage (at the cost of resource consumption) over MultiQueue which is constrained by single-queue access per instruction. This poses the question of whether MultiQueue is more of a trade-off and less of a direct improvement over existing architectures. This point should be made more clear. 
  • It is stated in the final paragraph of section 2 that the design attributes can be improved by optimizing 3 listed design parameters. These parameters are then assigned values in section 5, but with no justification as to if and why the chosen values are optimal, or how they were chosen. 
  • Testing is done using 755 instructions divided in 3 equal sets of each supported instruction. It is not stated whether the instructions are issued in a mixed sequence. Even though it is unlikely that the performance of the system would be affected, the test results could be presented with more confidence by using a realistic instruction sequence.  
  • Minor grammar and syntactic improvements are needed, such as  
  1. Line 48: … is in domain of networking, ... 
  1. Line 260: This is an typical way ... 

etc. 

Author Response

- there is no mention of energy performance and most importantly of throughput.  
=> added energy consumption comparison in the end of Section 4
- It is stated in the final paragraph of section 2 that the design attributes can be improved by optimizing 3 listed design parameters. These parameters are then assigned values in section 5, but with no justification as to if and why the chosen values are optimal, or how they were chosen
  => statement in Section 2 is corrected
  => parameters selection in Section 5 is explained
- the test results could be presented with more confidence by using a realistic instruction sequence.  
  => any instruction sequence works for any of the solutions mentioned in the paper - added explanation in Conclusion

Reviewer 3 Report

The paper is very similar to the author's previously published papers.

The novelty of the proposed architecture is not significant. authors use the random access feature of SRAMs to access elements in the priority queues. From the introduction, I expected multi-priority-queue management other than proposing the implementation of queues on SRAMs vs. LUTs.

To me, the authors require adding more details related to the management of the priority queues.

Author Response

reviewer #3:
- The paper is very similar to the author's previously published papers.
  => corrected a majority of paper rewording the text, redrawing Figure 1 and 2
  => the proposed solution (MultiQueue) was never published yet

Reviewer 4 Report

In this paper, the author presents a new priority Queue Architecture in FPGA. The paper is well written and easy to read. Although the paper is interesting, I can see some issues.

1) The images are very low quality and are blurred in my 27'' monitor. Also, there are no port names, just some lines. Some captions are not on the same page as the figure that corresponds.

2) The measurements are very limited. For example, the author presents Table1, but it does not specify (in the table) what the numbers mean. Furthermore, to evaluate a hardware design a lot of metrics should be present. Latency, Throughput, Area, and Energy consumption. Also, the experiments should be performed for a variety of boards not only for a single board. I would expect at least for Intel/Altera and Xilinx boards to see some results, and especially for the last architectures. Cyclone V is not one of the last architectures. the author should include more metrics, more Figures, more Tables to showcase the usefulness of his approach.

3) The contribution is very limited. In the 10 page manuscript, the contribution is 1 page and 1/2. This contribution is the mixing of two existing contributions. This could be publishable for a conference article, but for a journal, it is very limited. The authors should include more contributions (perhaps more architectures).

Author Response

reviewer #4:
- The measurements are very limited. For example, the author presents Table1, but it does not specify (in the table) what the numbers mean. Furthermore, to evaluate a hardware design a lot of metrics should be present. Latency, Throughput, Area, and Energy consumption. Also, the experiments should be performed for a variety of boards
  => the meaning of all columns were explained right above the table, and the attributes were explained in Section 2
- The contribution is very limited.
  => the proposed solution (MultiQueue) was never published yet, and it is providing significantly better resource utilization than the other solutions

Reviewer 5 Report

This paper describes a design for priority queues based on the shift register, systolic arrays, and SRAMs. There are some issues in this paper.

1. Some abbreviations used for the first time should be explained. For example, in line 15, universal verification methodology (UVM); In line 47, earliest deadline first (EDF) algorithm.

2. Some types, for example, in line 291, it should be FPGA Cyclone V.

3. In line 338, “The ALM consumption is much more important than SRAM bits. Most of digital system designs are typically limited by the ALM resource limitations, not SRAM bits” – This is not correct. For example, in deep learning applications, the RAMs are always the bottleneck instead of the registers. Even though the structures of registers and RAMs in FPGA are different, essentially both of them are used for storage. The difference is that registers can be accessed parallel, but the block RAMs can only access once per cycle depends on the data address. So you cannot ignore the usage of SRAM in the design. The comparison of ignoring the SRAM is unfair to other designs. 

4. In line 367, the number should be 1,398,784 instead of 1,284,672. And the numbers should be listed in the format such as 37,040 instead of 37040.

5. For the comparison results, the author only listed the resource usage, how about the speed and power which are also very important factors.

6. A 10 pages paper with 6.5 pages of paper introduction and background, only 3.5 pages related to the proposed MultiQueue, the main part listed on page 7 is very unclear. The author should strengthen the main part of the paper. Overall of the paper look like a report, but not a journal paper. The contributions and experiments are not enough, and most of the contents are shown up in previous papers.

Author Response

reviewer #5:
- The comparison of ignoring the SRAM is unfair to other designs.
  => the SRAM consumption is included in the results as well, depending on total system resource utilization, one can decide which solution is the most suitable (i.e. if someone has a lot of spare ALMs and not SRAM bits, one can decide to use a solution that does not use SRAM at all), but usually, ALMs are more critical in most systems/applications
- For the comparison results, the author only listed the resource usage, how about the speed and power which are also very important factors.
  => added speed and power comparison at the end of Section 4
- The author should strengthen the main part of the paper. Overall of the paper look like a report, but not a journal paper. The contributions and experiments are not enough, and most of the contents are shown up in previous papers.
  => that is probably due to the amount of solutions that were described, implemented and compared. But MultiQueue is a novel solution that was never published so far.

Reviewer 6 Report

In this paper the authors proposes an  SRAM-based architecture of a data structure that represents a set of multiple priority queues that can be implemented in FPGA or ASIC.

The introductory chapter is too general and the bibliographic references cover a very wide area and are quite vehement for this technology.

Chapter 3 describes architectures from the bibliography where are added figures without content.

 The chapter 4 where is expected to describe the work, it is very summary and the figures also are very generally.

Due to the very brief description of the proposed architecture, I consider the results are inconclusive. No correlations of the results with other works in this subject are made.

Author Response

The references cover wide area because the priority queues are used very widely in various applications.

Figure 1 and 2 were redrawn to be more clear. The text was rewritten too.

The Figure 5 and 6 are high-level intentionally - to avoid getting lost in less important details and because this approach is universally applicable to any set of priority queues, regardless of whether they are based on Shift Registers, Systolic Array or any other architecture.

Round 2

Reviewer 1 Report

The authors have done a lot of work to improve the quality of the paper. However the main weaknesses of the paper remain the same. In particular, Sections 4 and 5 which present the work and the results are the weakest sections.

A detailed block diagram is still missing in Section 4. Providing a high-level description of a HW design is not sufficient. For example Figure 5 and 6 do not show where the shared logic is and how it is accessed. There is just a MUX in Figure 5 which somehow gets results from the queues. There are also many confusing points which make the reader believe that the implementation is incorrect and it is not described in detail in purpose. For example, do the authors assume that the length (i.e. the number of registers) of each queue is the same? Is the number of the SRAM memories in Figure 6 equal to the queue length? If so, if we have really long queues does this mean that we need too many memories? 

Section 5 does not provide any significant performance results. We would expect to see the impact of the shared logic to the performance and the power. If I understand correctly the testbench was designed only to verify the design and not to get any performance results. If so, the testbench is straight forward and there is no need to explain it in detail.    

Author Response

Figure 5 is high-level because it is universal approach for any priority queue implementations. But we added more description in the text, explaining this and explaining the MUX's purpose/usage.

Thank you for very good questions, they are addressed in the new text in Section 4. Figure 6 was improved, adding more details + more text explaining the MultiQueue architecture and answering your questions. 

Section 5 is about testing whether the proposed solution is working as intended. Performance results were added and explained in Section 6 instead. All the solutions we are comparing are using a shared interface with insertion/deletion of one item in one queue at a time. Therefore the performance remains the same for all the solutions. The difference is in chip area / LUTs consumption only, but with a significant improvement.

Reviewer 3 Report

Thank you for addressing the issues. The paper sounds acceptable to me. Minor English revises are suggested.

Author Response

Thank you.

Reviewer 4 Report

In this revised version the authors have made some improvements but I believe is it not suitable for publication yet.

1) There is a lack of motivation. Even in the abstract, the author did not state why the MPQ is useful. In the introduction, the author gives some generic statements without backing them up with references. For example " in some cases, it is necessary to either sort/queue more data at a faster
rate or guarantee a constant response time" Here a reference is required, but none is given.

2) I find the author's response to reviewer comments not well stated and lacking all necessary details. The PDF uploaded with all the revision changes visible is very difficult to read. The author should have uploaded two versions, one with the revision markings and one with the last camera-ready version. Also, now the paper seems to be 15 pages, but in reality, it is much smaller.

3) the first 10 pages of the paper present the existing structure (related work). It looks like a survey paper but with a limited number of pages. From page 10 the author presents the architecture. The generic architectural figures that he presents are not sufficient to give all details. In just 1 and a half pages, the architecture is briefly explained, which is unacceptable for an architectural paper. How can this be verified?

4) The results are not well explained and could be biased towards the author's implementation.  The author wrote a couple of sentences for performance and for energy consumption, but he did not elaborate. the lack of details is very sketchy.

Overall, I believe strongly that this is not suitable for journal publications. lack of details, lack of discussion and analysis, and other omissions either make me think that there are serious gaps in this research, or the author wrote the manuscript in haste without being careful. I advise the author to read other architectural papers first and respect the audience and reviewers providing all necessary details in the manuscript, in order for the research to be easy to duplicate and test independently. 

Author Response

Thank you for your feedback, we tried to address your comments to improve the paper.

1) Priority queues are used in many applications ranging from various scheduling algorithms (task scheduling, network scheduling, etc.), graph algorithms (e.g. Dijkstra's algorithm), memory management (e.g. Worst-case algorithm) and many more. While normal systems can be OK with software implementation of priority queues, the real-time systems / safety-critical systems may have such timing requirements that might not be satisfied by software. Here comes hardware acceleration that can speed-up priority queues, but at a cost of relatively big amount of logic resources / chip area. The purpose of this research is to improve the chip area / resource costs - minimizing the main disadvantage of the HW approach and maintaining the same performance as long as only one priority queue is used at a time. Two more references to TTEthernet were added - these papers describe Time-Triggered Ethernet, which is implemented by HW switches on FPGA, which are using several priority queues for network scheduling. This is just yet another example of possible usage of the proposed MultiQueue solution.

2) the new uploaded zip file now contains 2 PDFs - one with marked changes (to see what was changed or added) and one without markings (the final version)

3) More details were added in Section 4 describing the MultiQueue architecture. The level of abstraction we used was selected to focus on core idea of the proposed solution. Adding too many details might make the readers to "get lost" in these details. Please let us know what details are missing from your perspective.

4) Performance and power consumption results were added in Section 6. The main point is that the proposed MultiQueue solution provides much better chip area / LUTs consumption results without any performance penalisation - provided that the original solution was using a single common (shared) interface allowing / needing only one instruction for one queue at a time anyway. We also highlighted limitations of the proposed solution - the number of priority queues and the common (shared) interface for all queues.

Reviewer 5 Report

The author solved some of the problems, but there are other problems still need to be addressed.

1. In line 421, it should be FPGA Cyclone V instead of “FPGA Cycle V”.

2. In line 521, systolic array solution consumed 1284672 ALMs which is not consistent with the data in Table 1. According to Table 1, the corresponding number should be 1398784.

3. For the result of bandwidth and power consumption, the qualitative description is not professional. Please list the quantitative numbers. For example, for the bandwidth, please list the numbers in the format such as bits/second or Mbits/second, or Mbytes/second. For the power, please list the value in Watts or mW. Otherwise, it is difficult to evaluate the performance.

Author Response

Thank you for your comments and feedback.

  1. - corrected
  2. - corrected
  3. - corrected - qualitative results of performance in Gbits/second and power analysis results in mW were added in Section 6. I would like to highlight that the proposed solution is better only in terms of chip area / LUTs resource consumption. The performance is the same as for multiple priority queues accesed with a MUX and the power results on FPGA are the same too (because same FPGA chip with same dynamic activation of logic is used. The saved logic was not active).

Reviewer 6 Report

The author answer to my observation.

Author Response

Thank you.

Round 3

Reviewer 1 Report

The manuscript has been significantly improved. The authors have clarified several technical questions. It is now clear that the proposed approach has several important weaknesses: large number of SRAMs, same queue capacity, number of queues should be a power of two. These weaknesses are now explained in the text. I think the impact of this paper is low but this is inherent in the proposed approach, so at this stage the authors cannot do much to improve the paper.

Author Response

Thank you for your review and feedback.

We tried to further improve the paper (changes are highlighed by red colour).

Yes, the number of SRAMs is equal to the priority queue depth and the number of queues should be a power of two. These limitations are causing that the proposed solution is not always suitable, highly depending on the requirements of the application. But if the application / system needs a lot of priority queues, where one is used at a time, then this proposed solution can be much better than any of the existing solutions because in most of the applicatons, the LUTs (ALMs) are needed/consumed much more than SRAMs - at least in data sorting applications.

Reviewer 4 Report

In this revised version, the author did improve it, but there are still issues that should be addressed.

First of all, the manuscript has too many bullets, making seem like a PowerPoint presentation and not like a consistent research paper. The authors should try to utilize another presentation method, more suitable for a document. This is a minor comment.

Figure 1 is similar to Figure 2 with only a name change. What is the purpose of including an almost similar Figure? The authors should include more details on the Figures to illustrate the different architectures.

The first half of the paper provides a good survey of the techniques and should remain on the manuscript.

A major remark is that: Once more, the architecture is not clearly described. After reading this I can say that I cannot utilize this knowledge to design a MultiQueue. I propose the author include an MWE (minimum working example) for some Bytes to give

the full details of the design process.

Author Response

Thank you for your review and feedback.

We tried to improve the paper in all aspects you mentioned.

The number of bullets was reduced - some were replaced with plain text, some with a table.

Figure 2 had a mistake - we updated this diagram with a newer version that is more accurate. Also, we added more text explaining the difference between these two solutions.

We added more details about the proposed solution. Figure 6 was updated by adding Sorting Cells within the architecture and by adding arrows/interfaces between the components. We also added a new diagram (Figure 7), which describes the internal structure and behavior of the logic inside Sorting Cell. We also added more text (2 paragraphs) describing the proposed solution - one at higher level (describing the architecture - page 8) and one at lower level (describing the Sorting Cell - page 9).

Round 4

Reviewer 4 Report

The paper has been improved and I believe it can be published in this form.